# Characterizing the Mechanism of Action of Essential Oils on Skin Homeostasis—Data from Sonographic Imaging, Epidermal Water Dynamics, and Skin Biomechanics

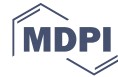

Sérgio Faloni de Andrade [1], Patricia Rijo [1], Clemente Rocha [1], Lin Zhu [2] and Luis Monteiro Rodrigues [1,*]

1   CBIOS—Research Center for Biosciences and Health Technologies, Universidade Lusófona, Campo Grande 376, 1749-024 Lisboa, Portugal; sergio.andrade@ulusofona.pt (S.F.d.A.); patricia.rijo@ulusofona.pt (P.R.); c.rocha.dr@gmail.com (C.R.)
2   Jiangxi University of Traditional Chinese Medicine, Walin, Nanchang 330000, China; 13130173400@163.com
*   Correspondence: monteiro.rodrigues@ulusofona.pt; Tel.: +351-217-515-500

**Abstract:** Essential oils (EOs) have been recognized as materials of interest for dermatological applications, although some doubts remain regarding their safety and efficacy. We studied the action mechanisms of EOs from lavender and sage in human skin. Extracted EOs were incorporated (at 5% and 10%) in almond oil as a vehicle. Eleven healthy volunteers were selected and the prepared oils were tested on both forearms. All procedures respected the principles of good clinical practice. Effects were followed through high resolution sonography (HRS), epidermal water dynamics, and biomechanics. All variables were measured before and 30 min after application. Nonparametric statistical comparisons were applied ($p < 0.05$). HRS revealed a more echogenic epidermis, with a significant echogenicity decrease in the dermis (higher water retention) for all formulations. Significant TEWL decrease and an increase in superficial and deep epidermal hydration were also observed. These results indicate that EOs penetrate only into the most superficial layers of the skin, which is important for their safety profile. Furthermore, this "filmogenic" mechanism improving the epidermal water balance seems to connect directly with the observed biomechanical enhancement. These results confirm the clinical relevance of these compounds, in particular to restore the epidermal water content and prevent xerosis and other related disorders in sensitive (atopic, elderly) patients.

**Keywords:** essential oils; mechanism of action; sonography; evaporimetry; epidermal water dynamics; skin biomechanics



## 1. Introduction

Essential oils are complex mixtures of volatile low molecular weight compounds (monoterpenes and sesquiterpenes) responsible for the characteristic aroma of each plant. Some of these complex organic combinations, extracted by aqueous distillation, have attracted attention from the industry (from chemical, perfume and cosmetics to food) where these oils are already used for their aromatic properties.

More recent interest from the pharmaceutical industry is due to scientific studies reporting a wide potential of beneficial activities, including antioxidant, antimicrobial, anti-inflammatory, and even anti-neoplastic properties [1–4]. Moreover, several essential oils and related lipidic compounds have been incorporated into nanostructured systems for the preparation of over-the-counter formulations for many different purposes [5].

However, some reported adverse effects (mucosal irritation, pain, diarrhea) and chronic toxicity (in human skin cells) raised justified concerns that have limited their development and application in human research [6,7] and recommended dilution for any (internal as external) usage to reduce the exposure risk [8]. Several publications have demonstrated a good safety pattern for some of these compounds, especially for topical use, a major application area of interest [9,10]. A recent paper identified approximately

1500 combinations of 90 essential oils with dermatological interest [11]. The increased use of plant oils within skincare formulations may result, at least in part, from the observed activity of their constituents to contribute to cutaneous lipid balance and therefore to preserve or to reestablish the skin epidermal "barrier", although doubts still exist regarding their penetration capacity and mechanisms of action [12,13].

The present study is focused on the identification of the biological impacts of *Lavandula angustifolia* Mill. (lavender) and *Salvia officinalis* L. (sage) essential oils. These plants from the Lamiaceae family originated in the Mediterranean and are abundant in the Portuguese territories. Both plants are a major source of essential oils, and have long been used in traditional medicine as in modern phytotherapy [14]. The oil from *L. angustifolia* is known for its high content of linalool and linalyl acetates and has been recommended for wound treatment, eczema, and psoriasis [14,15]. *S. officinalis* and its essential oil are rich in alpha- and beta-thujone, camphor and 1,8-cineole, and have been used to treat upper GI tract inflammation and infection [16,17].

Our study involved the topical application of 5% and 10% essential oils extracts from lavender and sage to healthy participants after confirming adequate safety and skin tolerance profiles. Following application, skin was analyzed using high-resolution sonography imaging coupled with functional measurements of epidermal water dynamics and biomechanics in order to better understand the mechanism of action and the potential clinical relevance of these compounds.

## 2. Methods

### 2.1. Participants

Our sample involved 11 healthy individuals, both sexes (5 men and 6 women), aged between 18 and 45 years old (mean 31.3 ± 10.0). Criteria for participation included the confirmation of (i) no visible cutaneous lesions and no past or present record of dermatological disease or atopy, (ii) absence of the application of any cosmetics in the test area 48 h prior to the study at baseline, as well as (iii) any pharmacological treatment that might interfere with measurements, and (iv) no recent sun or solarium exposure prior to the study.

All procedures observed the principles of good clinical practice from the Helsinki Declaration and respective amendments [18], including an informed written consent. The study was formerly approved by the Institutional Ethical Committee.

### 2.2. Essential Oils Obtention and Formulations

The essential oils were extracted from 52.5 g (g) of commercially available dry samples of *Lavandula angustifolia* flower (Alfazema, Celeiro, Lisboa, Portugal) and *Salvia officinalis* leaves (Salvia, Celeiro, Lisboa, Portugal). Plants were hydrodistilled for 2 h using a Clevenger's apparatus. After hydrodistillation, 1.1 mL of *L. angustifolia* essential oil was obtained, corresponding to a yield of 2.1% (*w/v*), while for *S. officinallis,* 0.28 mL of essential oil was obtained, corresponding to a yield of 0.56% (*w/v*).

Both essential oils obtained were diluted in almond oil (F.J.Campos, Lisboa, Portugal) to final concentrations of 5% (*v/v*) and 10% (*v/v*) for use in the experimental study.

### 2.3. Procedure

Our experimental design was patterned from previously published studies with comparable ingredients and purposes [12] preceded by a "primary irritancy test" to confirm the safety of use of the diluted forms (5% *v/v* and 10% *v/v*) of both EOs.

All measurements were made in a humidity- and temperature-controlled environment (humidity ~50%; temperature 21 ± 2 °C), to which participants were acclimatized for 30–60 min prior to the evaluations. Six sites (3 cm × 3 cm) were marked in the ventral aspect of each participant's right and left forearms. Each of the four prepared extract solutions (5% and 10% *L. angustifolia* and 5% and 10% *S. officinalis*) and the commercially available almond oil (negative control/dilutant) were randomly applied (Latin square),

while one of the sites was left empty under occlusion (parafilm covered by an adhesive patch) and used as a positive control. Formulations were applied (2 mg/cm$^2$) with a small spatula and left in contact with the skin for 30 min. After the elapsed time, occluded sites were uncovered for further measurements and the remaining oil, if any, was removed with absorbent paper.

We used various approaches to study the impact of the prepared essential oils on human skin in vivo before and 30 min after application. A bidimensional color image was obtained via high resolution sonography (HRS, Dermascan C, Cortex Technology, Hadsund, Denmark), with echo recorded at a velocity of 1580 m/s using a 20 MHz probe placed on the skin in a fixed standard position [19]. Using ImageJ® software (NIH, Bethesda, MD, USA), the color image was converted into grayscale and further analyzed.

Functional measurements included (i) the epidermal "barrier" function, (ii) epidermal hydration, and (iii) skin biomechanics. Epidermal "barrier" function was quantified as Transepidermal Water Loss (TEWL) by an evaporimeter system (Tewameter TM300 CK electronics GmbH, Cologne, Germany) and expressed in g/cm$^2 \cdot$s$^{-1}$ [20,21]. Epidermal hydration was measured using the MoistureMeter SC and MoistureMeter D systems (Delphin Technologies, Kuopio, Finland), both of which are "electrometrically" based, using different frequencies to provide superficial (SC) and deep hydration values (D), equally expressed in arbitrary units (AUs) [22]. Skin biomechanics were assessed with the Cutometer®MPA580 (CK electronics GmbH, Cologne, Germany) equipped with a 2 mm aperture probe, which exerts a controlled negative pressure on the skin surface. From this apparent stress-strain curve, different biomechanical "descriptors" can be quantified [23,24]. For our study, we selected those descriptors identified by Agache; that is, Uf (total elongation or R0), Ua (total recovery or R8) expressed in mm, and the ratios Ur/Ue (net elasticity or R5) and Uv/Ue (viscoelasticity or R6) as main indicators [24]. The experimental design is summarized in Figure 1.

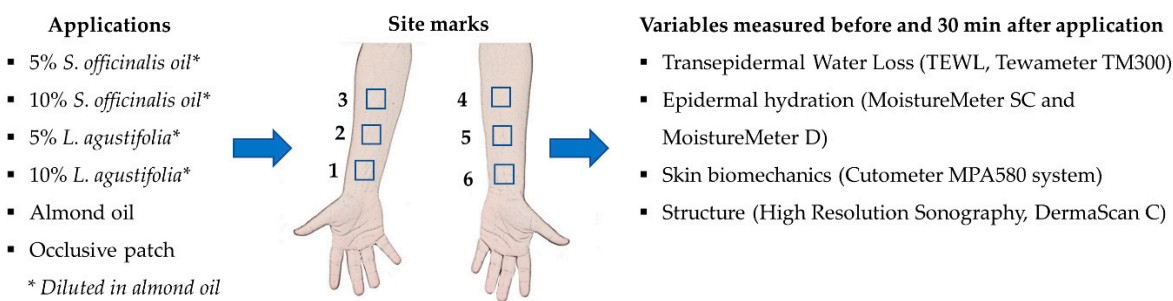

**Figure 1.** Graphical representation of the study design. Formulations (left) and measured variables (right) are indicated. Formulations (four) and controls (two) were applied to sites marked on both ventral forearms in a random sequence previously set for each volunteer (Latin square). Measurements took place in each site before and 30 min after applications (see Section 2.3 for details).

### 2.4. Statistics

Data reported as mean ± standard error of the mean (SEM) were compared by one-way analysis of variance (ANOVA), followed by Bonferroni's test or Wilcoxon signed-rank test using GraphPadPrism 5® software (GraphPad software, San Diego, CA, USA). A confidence level of 95% ($p < 0.05$) was adopted.

### 3. Results

The preliminary safety screen revealed a complete absence of skin reactions and thus an excellent tolerance to the diluted EOs as tested (data not shown). Table 1 summarizes data from the sonographic consequences of the application of the preparations on in vivo human skin. The present conditions used were adapted from a comparable reference study [12]. An illustrative example is shown in Figure 2. As shown, the epidermis is clearly more echogenic (brighter colors) after the application and 30 min exposure of all

preparations. In the dermis, however, a significant echogenicity decrease (darker contrast) is detected for all extracts, in particular for the 5% formulations, especially when compared with controls (negative—the "blank" almond oil and positive—the occluded empty site).

**Table 1.** Percentage variation of the echogenicity in epidermal and dermal areas 30 min after the application of the studied different formulations. Results are expressed as mean $\pm$ S.E.M. Statistical comparison, before and after the application, was performed using one-way ANOVA followed by Bonferroni's test. * $p < 0.05$, ** $p < 0.01$.

| Treatment | TEWL g/m²/h before (T0) after (T30) | | Epidermal Superficial Hydration (AU's) before (T0) after (T30) | | Epidermal Deep Hydration (AU's) before (T0) after (T30) | |
|---|---|---|---|---|---|---|
| EO *S. officinalis* 5% | 6.36 $\pm$ 0.61 | 4.43 $\pm$ 0.34 ** | 32.80 $\pm$ 2.78 | 41.51 $\pm$ 3.08 ** | 17.38 $\pm$ 0.92 | 20.04 $\pm$ 0.63 ** |
| EO *S. officinalis* 10% | 6.91 $\pm$ 0.73 | 5.14 $\pm$ 0.59 * | 33.55 $\pm$ 1.93 | 42.35 $\pm$ 2.73 ** | 18.01 $\pm$ 0.95 | 19.81 $\pm$ 0.69 * |
| EO *L. angustifolia* 5% | 6.50 $\pm$ 0.63 | 4.69 $\pm$ 0.51 ** | 33.88 $\pm$ 2.37 | 44.41 $\pm$ 4.64 * | 17.29 $\pm$ 0.85 | 19.18 $\pm$ 0.86 ** |
| EO *L. angustifolia* 10% | 6.68 $\pm$ 1.81 | 4.84 $\pm$ 1.70 ** | 32.12 $\pm$ 2.33 | 46.25 $\pm$ 3.67 ** | 17.86 $\pm$ 0.95 | 20.11$\pm$ 0.69 ** |
| Almond Oil | 5.99 $\pm$ 0.81 | 4.22 $\pm$ 0.55 * | 31.59 $\pm$ 3.16 | 40.61 $\pm$ 3.67 ** | 16.44 $\pm$ 0.81 | 18.48 $\pm$ 0.76 * |
| Occlusive Patch | 6.95 $\pm$ 0.80 | 10.62 $\pm$ 1.33 | 33.86 $\pm$ 2.22 | 42.50 $\pm$ 3.20 | 17.50 $\pm$ 0.62 | 18.69 $\pm$ 0.64 |

**Before**        **After**

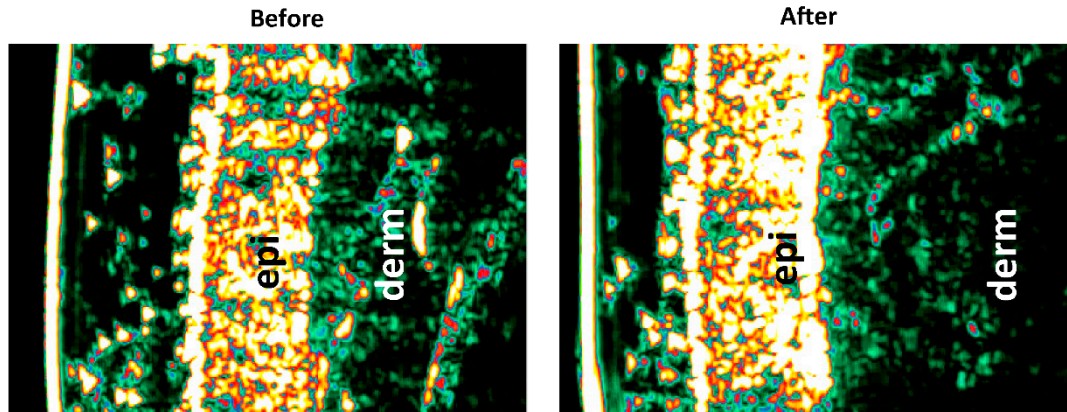

**Figure 2.** Illustrative example of the impact of an essential oil formulation (*S. officinalis* 5%) on the sonographic structure of one participants' skin. Images, obtained by HRS before and 30 min after the application, shows an increase in the echogenicity at "epi" (epidermal layer), while in the "derm" (dermis) area echogenicity clearly decreases (see text).

Table 2 summarizes data related to epidermal water dynamics. TEWL values were significantly decreased with the application of 5% and 10% dilutions of *S. officinalis* and *L. angustifolia* when compared with baseline. The same effect was observed for the almond oil used as vehicle (negative control). The occluded site (positive control) shows an opposite effect. Significant epidermal hydration changes were also detected in deeper epidermal layers ($p < 0.01$ for EO 5% *S. officinalis* and $p < 0.05$ for other dilutions) with the tested extracts. In contrast, almond oil and the occluded empty site show only superficial hydration changes.

Figure 3 summarizes the impact of those formulations on skin biomechanics. All essential oil extract preparations improved skin biomechanics in terms of (i) Uf (total elongation or R0), the maximum amplitude and quantitates the passive behavior of the skin to force (firmness), measured in mm; (ii) Ua (total recovery or R8), the maximum relaxation after suction is released, expressed in mm; (iii) Ur/Ue (net elasticity or R5), corresponding to total elasticity; and (iv) Uv/Ue (viscoelasticity or R6), corresponding to the relationship between elastic and viscoelastic extension. As shown, the most consistent and significant differences were observed with the 5% dilutions.

**Table 2.** Skin echogenic changes observed in vivo, at the test sites (%) registered 30 minutes (T30) after the application of the studied formulations, compared to T0 before intitiating the study. Data are expressed as mean $\pm$ SEM (n = 11). Comparative statistics comprises T30 to T0 for each variable (Wilcoxon) (* $p < 0.05$, ** $p < 0.01$).

|  | Echogenic Changes/Area (%) | Echogenic Changes/Area (%) |
|---|---|---|
|  | Epidermis | Dermis |
| EO *S. officinalis* 5% | 20.10 $\pm$ 1.70 | −27.81 $\pm$ 4.88 ** |
| EO *S. officinalis* 10% | 14.62 $\pm$ 1.60 | −22.63 $\pm$ 1.52 * |
| EO *L. angustifolia* 5% | 19.02 $\pm$ 2.70 | −28.53 $\pm$ 4.46 ** |
| EO *L. angustifolia* 10% | 15.95 $\pm$ 2.68 | −25.62 $\pm$ 4.95 * |
| Almond Oil | 14.34 $\pm$ 3.52 | −17.88 $\pm$ 3.70 |
| Occlusive Patch | 15.84 $\pm$ 0.72 | −7.25 $\pm$ 2.06 |

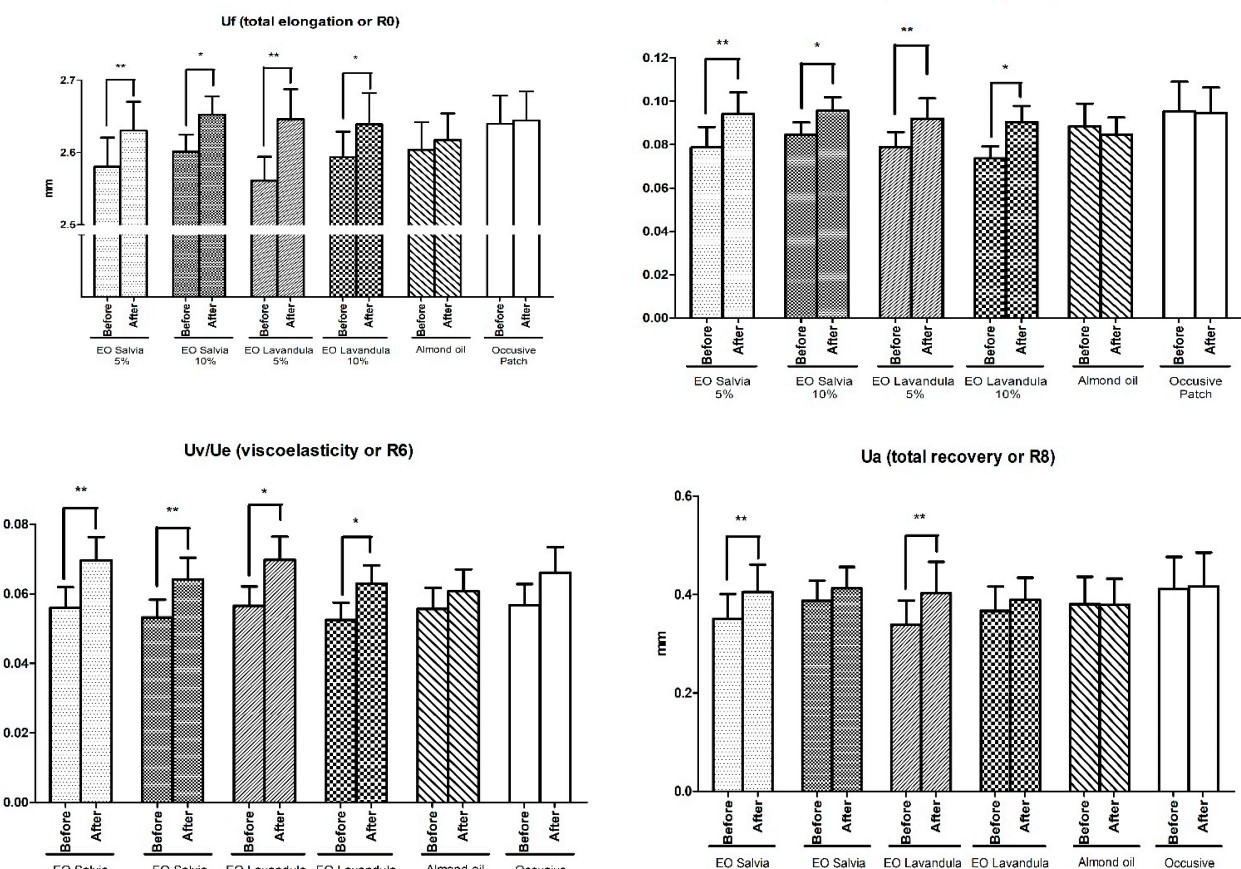

**Figure 3.** Skin biomechanical descriptors expressed by the indicated variables, before (T0) and 30 min (T30) after the application of the studied formulations. Data are expressed as mean $\pm$ SEM (n = 11). Comparative statistics equates T30 to T0 for each variable (Wilcoxon) (* $p < 0.05$, ** $p < 0.01$).

## 4. Discussion

Essential oils of plant origin have garnered remarkable interest from pharmaceutical and cosmetic industries far greater than their related (cold pressed) "fixed oils". There are considerable physical, chemical, and biological differences between these two classes of natural oils, as they are obtained by different processing and refinement methods, with very distinct yields that determine their specific composition and production costs [2].

For numerous reasons, plant essential oils have gained remarkable prominence in the skincare industry, with a significant increase in their application in capsules, syrups,

ointments, creams, and sprays. These forms extend from pharmaceutical to cosmetics to treat bacterial, fungal, and viral infections, as well as inflammation, acne, and many other skin disorders. Essential oils have been used as skin penetration enhancers for transdermal drug delivery, as well [10,25], and have gained particular prominence to address xerotic and inflammatory dermatoses affecting the epidermal skin barrier. The importance of a balanced content of intercellular lipids to preserve or repair the epidermal skin "barrier" is known, but only linoleic acid and oleic acids ratio have been recognized as important [26,27].

On the other hand, essential oils seem to be responsible for a higher incidence of sensitization reactions, including allergic contact dermatitis, than fixed oils [7,28]. Nevertheless, essential oils are economical and readily available, both of which are important arguments for being preferred, and many positive impacts of these compounds on human skin have been reported [29,30]. Comprehensive research on their in vivo human effects and mode of action has only recently begun to accumulate, and results are sometimes controversial. Recent reports underlined their superficial character, acting principally in the outer epidermis, while other studies suggest that their constituents can penetrate and even increase penetration of other substances, acting therefore as permeation enhancers of other active drugs [1,13,31].

An extensive investigation comparing the different penetration of several oils (jojoba, soybean, avocado, paraffin, almond) and petrolatum confirmed the distribution of these compounds into the superficial epidermis, excluding deeper penetrations while identifying a semi-occlusive effect based on the reduction of TEWL [12]. The in vivo application of sophisticated imaging systems, such as confocal Raman microspectroscopy or laser scanning microscopy, proposed that these oils typically penetrate into the upper epidermis, although more sensitive approaches, such as in vivo fluorescence microscopy, could detect these compounds at the dermal-epidermal transition [32–34]. Investigations with *Melaleuca alternifolia* essential oil (tea tree oil) at 5%, the most active concentration used in our study, have shown that skin layers contained (in total) less than 1% of each tea tree oil marker after application. Only oxygenated terpenes significantly permeated across the skin, while hydrocarbons were only absorbed at trace levels and substantial concentrations of many markers were released into the atmosphere [35].

Our main purpose was to explore the impact of diluted *S. officinalis* and *L. angustifolia* essential oils in the human skin. To the best of our knowledge, no similar studies regarding the safety and efficacy of these two oils have been published. Preliminary safety screens indicated excellent tolerance of these EOs. HRS imaging data (Table 1 and Figure 2) revealed an increase in the epidermal echo intensity, although not significantly different from the baseline measurements. However, major differences were noted at the dermis, where a dramatic reduction of the echo was observed for all EO preparations, higher at 5% than 10%. In our opinion, these two effects are related, meaning that a reinforcement of the epidermal cohesion, revealed by the increased epidermal echogenicity, reduces the water gradient from deeper tissues to the surface, promoting water retention at dermis and thus reducing its echogenicity. The almond oil and the occlusion evoked similar effects, but much more discrete (Table 1). These results agree with previous observations on this issue [12,32,34], aligning with the known cohesiveness reinforcement capacity of these oils and their lipid character on the stratum corneum and confirmed by the changes detected in the epidermal water dynamics (Table 2). TEWL was significantly reduced by all formulations after application, which is perceived as a reinforcement of the epidermal "barrier" efficacy against desiccation. This effect was previously noted [12] and confirms our views on the mechanism of action involved. As a consequence, significant water amounts were detected in the epidermis, especially in the deeper layers. The measurement depth of skin water varies with the electromagnetic field created by the systems' "capacitor" with different frequency waves—1.25 MHz for the Moisturemeter SC and 300 MHz for the Moisturemeter D. The calculation of the water dielectric constant is assumed to be proportional to the water content of the measured tissue considering the thickness of the

stratum corneum [36–38]. This mode of action—reducing the water loss and promoting its accumulation on the viable epidermis—seems to impact deeper skin layers, as suggested by the dermal echogenicity reduction previously mentioned (Table 1, Figure 1). Consequently, these changes also seem to favor skin biomechanics, although a clear relationship between skin water dynamics and biomechanics is not clear. A few studies have demonstrated a significant improvement in the descriptors of maximum extensibility elastic function and viscoelastic ratio related with the daily water intake [39,40]. Our data clearly show a significant improvement of all descriptors (Figure 3) following the short-term application of the extracts under study, more consistent with the water retention mechanism affecting the deepest skin structures rather than with a superficial effect on *stratum corneum*. While the relative participation of the epidermis in skin biomechanics is still unclear, the contribution of the tissues underneath has been established [25,41]. The occlusive capacity of these substances on the epidermis provides sufficient evidence of deep water retention capable of improving the biomechanics of the skin. Furthermore, these results confirm and reinforce the therapeutic interest of these compounds in dermatology and skincare. Keeping in mind the recognized importance of epidermal water balance to preserve the "barrier" function and the growing incidence of xerosis, pruritus, and skin irritation in elderly populations, these compounds might be logical additions to pharmaceutical and cosmetic formulations for the treatment and prevention of dry skin, atopic predisposition, or even occupational diseases.

## 5. Conclusions

To the best of our knowledge, this is the first study focusing on the biological impact and mechanism of action of *S. officinalis* and *L. angustifolia* essential oils on human skin in vivo. These oils were applied in dilute concentrations in an effort to prevent any potential adverse reaction, as they are used in traditional medicine (e.g., massage) and modern phytotherapy. Our study demonstrated that these dilutions penetrate only the most superficial layers of the skin, confirming the safety of their use while promoting a statistically significant improvement of various cutaneous properties, including the reinforcement of the epidermal barrier, deep hydration, and biomechanical behavior. These effects were more pronounced with both essential oils at 5%, and most were more pronounced than the pure almond oil vehicle. Similar observations were previously reported [29]. Finally, it is noteworthy that these benefits were observed in simple (oil) dilutions, an advantage in formulation preparation and expanding the potential use of these oils in skin health.

These evidence of safety of use and beneficial effects on human skin in vivo, even in such a reduced concentration, confirms the interest of these essential oils for the development of dermatological formulations to improve skin health and well-being.

**Author Contributions:** P.R. prepared the formulations; C.R., L.Z., and S.F.d.A. assumed the in vivo experimental procedures; S.F.d.A. and L.M.R. wrote, corrected, and approved the manuscript in its final form. All authors have read and agreed to the published version of the manuscript.

**Funding:** This research is funded by Fundação para a Ciência e a Tecnologia (FCT) through grant UIDB/04567/2020 to CBIOS. Sérgio Faloni de Andrade is funded by Foundation for Science and Technology (FCT)—Scientific Employment Stimulus contract with the reference number CEEC/CBIOS/PMHD/2018.

**Institutional Review Board Statement:** Not applicable.

**Informed Consent Statement:** Informed consent was obtained from all subjects involved in the study.

**Data Availability Statement:** Data supporting reported results can be requested directly to the corresponding author.

**Acknowledgments:** The authors acknowledge all participants and Henrique Silva.

**Conflicts of Interest:** The authors declare no conflict of interest.

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
