# Peer review of "Characterizing the Mechanism of Action of Essential Oils on Skin Homeostasis—Data from Sonographic Imaging, Epidermal Water Dynamics, and Skin Biomechanics"

_cosmetics, doi:10.3390/cosmetics8020036_

Round 1

Reviewer 1 Report

In Introduction the authors should mention the use of nanotechnologies applied to cosmeceuticals and essential oils

A graphical rapresentation of study design should be inserted.

Table 1 should be checked and reorganized.

Data in Figure 2 should be better described in the text.

Limits, asvantages and practical application should be inserted in the Conclusion.

Author Response

Comments and Suggestions   

  1. In Introduction the authors should mention the use of nanotechnologies applied to cosmeceuticals and essential oils

REPLY: Done – lines 36-45, an additional reference on this subject was included

  1. A graphical representation of study design should be inserted

REPLY: Done as suggested and included as Figure 1

  1. Table 1 should be checked and reorganized

REPLY: Table 1 was fully checked and reorganised

  1. Figure 2 should be better described in the text

REPLY: Done – lines 156-160; note that Figure 2 is now Figure 3.

  1. Limits, advantages and practical application should be inserted in the Conclusion

REPLY: The conclusion section was re-written (lines 298-308) for clarity, considering the reviewers’ remarks

Reviewer 2 Report

Dear authors

Essential oils (EOs) have been recognized as materials of interest to the industry for dermatological applications. The extracted EOs were incorporated into almond oil as a vehicle. Eleven healthy volunteers were selected and tested in both forearms. All procedures respected the principles of good clinical practice. The effects were followed up through high resolution ultrasound and quantification of epidermal water dynamics and skin biomechanics. The authors applied non-parametric statistical comparisons (p <0.05). HRS revealed a more echogenic epidermis, with a significant decrease in echogenicity in the dermis, greater water retention, for all formulations. A significant decrease in TEWL and a significant increase in superficial and deep epidermal hydration was also observed. Therefore, EOs only penetrate the most superficial layers of the skin, which is important for their safety profile. Furthermore, this "filmogenic" mechanism that improves epidermal water balance seems to connect directly with the biomechanical improvement observed. These results confirm the clinical relevance of these compounds in particular for restoring epidermal water content and preventing xerosis and other related disorders in sensitive patients (atopic, acne).

Analisi of Paper :

1 - Introduction: it must be reformed in the content and in the writing of the general part to review the syntax of the topic

2- Discussion: deepen in consideration of the problem of drugs on the market with poor efficacy in inflammatory pathologies and also of infectious origin, essential oils have a good penetrating power and are very effective in restoring skin hydration. It has been seen to be very effective in xerosis, atopy and acne diseases and therefore good use in the future. Learn more about this topic using and citing the following references:

PMID: 32210603 ; PMID: 30039757 ; PMID: 32097972 

3 - Check the bibliographic entries throughout the text, some of which are non-compliant, review some entries in the bibliographic references and necessarily insert those referred to in point 2 for the purpose of acceptance by me.

4 - Review the English grammar and in particular the applied scientific English: in particular, the verb tenses and the syntax in the discussion.

Author Response

  1. Introduction: it must be reformed in the content and in the writing of the general part to review the  syntax of the topic In Introduction     

REPLY: We have reviewed the complete section for clarity, having in mind all the comments received. We believe the text is now more objective and concise

  1. Discussion: deepen in consideration of the problem of drugs on the market with poor efficacy in inflammatory pathologies and also of infectious origin, essential oils have a good penetrating power and are very effective in restoring skin hydration. It has been seen to be very effective in xerosis, atopy and acne diseases and therefore good use in the future. Learn more about this topic using and citing the following references: PMID: 32210603 ; PMID: 3003975 ; PMID: 32097972

REPLY: Done. We have extended the discussion to these issues and included the suggested references into the text and reference list

  1. Check the bibliographic entries throughout the text, some of which are non-compliant, review some entries in the bibliographic references and necessarily insert those referred to in point 2

REPLY: Done – all entries were re-assessed in order to confirm the references in the text, including those suggested by the reviewer.

  1. Review the English grammar and in particular the applied scientific English

REPLY: Done. The final text was reviewed by an experienced, native (American) English science editor.

Reviewer 3 Report

The manuscript by de Andrade et al. explores the possible effects of essential oils on the skin homeostasis. The work is well performed and the results are sound. I have to minor points that the authors should address before acceptance:

-The multiple applications of the essential oils is clear. However, from the manuscript It is not clear the cosmetic interest of essential oils.

-may the data be extrapolated to higher concentration of essential oils? Essential oils are good solvent of different hydrophobic molecules and It can be interesting to use higher concentration. Some discussion should be included.

-are differences expected for encapsulated essential oils? This is the most common way in which essential oils are used

Author Response

Comments and Suggestions

  1. … from the manuscript It is not clear the cosmetic interest of essential oils.

REPLY: The Introduction and Discussion sections were fully revisited to meet and clarify this aspect. 

  1. may the data be extrapolated to higher concentration of essential oils? Essential oils are good solvent of different hydrophobic molecules and It can be interesting to use higher concentration. Some discussion should be included

REPLY: As mentioned, we used EO dilutions (mineral oil) and obtained more consistent results from 5% rather than with 10% concentration. This observation was consistent with previously published results; however, a potential explanation is out of the scope of the present paper. Nevertheless, these aspects are discussed in the respective section.

  1. are differences expected for encapsulated essential oils?

REPLY: We understand the curiosity of the reviewer, and it is an interesting avenue to consider. Our aim, however, was centered in the mechanism of action of a simple (and fast) O/O dilution. We believe we fully achieved that objective showing the effectiveness of these oils under these conditions. In our opinion, other extrapolations should not be made from these results. 

Round 2

Reviewer 2 Report

Corrections made correctly.